# Conditional Adapters: Parameter-efficient Transfer Learning with Fast Inference

**Tao Lei**[*]  **Junwen Bai**  **Siddhartha Brahma**  **Joshua Ainslie**  **Kenton Lee**  **Yanqi Zhou**
**Nan Du**  **Vincent Y. Zhao**  **Yuexin Wu**  **Bo Li**  **Yu Zhang**  **Ming-Wei Chang**

Google

## Abstract

We propose Conditional Adapter (CODA), a parameter-efficient transfer learning method that also improves *inference efficiency*. CODA generalizes beyond standard adapter approaches to enable a new way of balancing speed and accuracy using conditional computation. Starting with an existing dense pretrained model, CODA adds sparse activation together with a small number of new parameters and a light-weight training phase. Our experiments demonstrate that the CODA approach provides an unexpectedly efficient way to transfer knowledge. Across a variety of language, vision, and speech tasks, CODA achieves a 2x to 8x inference speed-up compared to the state-of-the-art Adapter approaches with moderate to no accuracy loss and the same parameter efficiency.

## 1   Introduction

Large pretrained models have achieved groundbreaking results but the main impediment to deploy them has been the cost of adaptation and inference. Due to the ever growing size of the pretrained models, for example, finetuning has become increasingly expensive as it requires a separate copy of the full model and updates to all parameters for every downstream task. Parameter-efficient transfer learning such as Adapter [Houlsby et al., 2019] and Prompt Tuning [Lester et al., 2021] have been proposed to address this issue. These methods only update a small subset of parameters for each downstream task, allowing the model to retain knowledge and avoid catastrophic forgetting [Vu et al., 2022]. Noticeably, these methods can match the accuracy of a fully finetuned model, while achieving better accuracy on out-of-domain data distributions [Lester et al., 2021, Awadalla et al., 2022].

Unfortunately, standard parameter-efficient transfer learning methods only bring *parameter* efficiency, not *inference* efficiency. For example, while only a few small projection matrices are added into the pretrained model in the Adapter approach, all the model inputs (such as tokens) still use all parameters during inference. Therefore, the inference speed is the same (or slightly lower) with respect to the full finetuning method. Moreover, prior studies have shown that these parameter-efficient learning methods are most effective when the size of the pretrained model is large [Lester et al., 2021], making many advantages of these methods difficult to realize in practice.

In this paper, we propose Conditional Adapter (CODA), a parameter-efficient transfer learning method that offers both *parameter* and *inference* efficiency. CODA is a generalization of the adapter approach, built with the following intuition – we can treat the pretrained model as a universal source of knowledge but only query against it for *necessary inputs*. Figure 1 compares CODA with finetuning and standard adapter approaches. Similar to standard adapter approaches, our model adds and updates a small adapter in each layer, while fixing the pretrained Transformer blocks for downstream adaptation. Unlike previous approaches, however, CODA assumes that many of input

---

[*]Correspondence: `taoleics@gmail.com`, `junwen@google.com`

37th Conference on Neural Information Processing Systems (NeurIPS 2023).

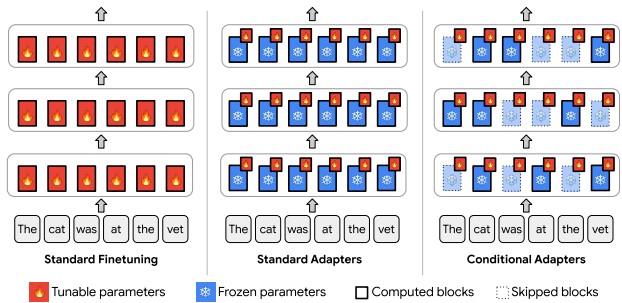

| | New param | MNLI (text) | |
|---|---|---|---|
| | | Acc ↑ | Speedup |
| P-Adapter | 0.4% | 91.5 | 1.0x |
| CoDA | 0.4% | 90.7 | **3.2x** |
| | New param | OCR-VQA (vision) | |
| | | EM ↑ | Speedup |
| P-Adapter | 2.8% | 67.5 | 1.0x |
| CoDA | 2.8% | 67.6 | **8.0x** |
| | New param | Librispeech (speech) | |
| | | WER ↓ | Speedup |
| P-Adapter | 2.5% | 1.4/2.7 | 1.0x |
| CoDA | 2.5% | 1.4/2.8 | **2.2x** |

Figure 1: Comparison between different ways to use pre-trained Transformer models, including (1) standard finetuning (left) where all parameters are tunable and computation is dense, (2) standard adapters (center) where a small set of new tunable parameters are added while the computation remains dense, and (3) CoDA (right) where the computation is sparsely activated.

Table 1: CoDA significantly reduces the inference time compared to the Parallel Adapter approach [He et al., 2021], while still maintaining parameter efficiency.

token representations (of each layer) are not important for the prediction task and therefore do not require heavy computation. In such cases, the pretrained Transformer block can be skipped. Given that many tokens are not processed by the Transformer block, CoDA runs significantly faster than previous methods.

While conditional activation has clear speed benefits, CoDA must learn to select important tokens for heavy computation in order to maintain model accuracy. To this end, we introduce a soft top-$k$ operation for computing the token selection decision. This soft top-$k$ operation, which can be seen as a generalization of softmax and a relaxation of hard top-$k$, utilizes entropy-regularized optimization techniques similar to computational optimal transport [Cuturi, 2013]. As a result, its output can be computed using fast and differentiable iterations, allowing token selection to be directly optimized for model performance.

We apply CoDA on encoder-heavy tasks and evaluate its effectiveness on three different domains – natural language processing, computer vision and speech processing. Overall, CoDA achieves 2 to 8 times inference speed-up over standard adapter approach with moderate to no accuracy loss. Table 1 showcases our results by selecting one of the best performing tasks in each domain. We also conduct comprehensive ablation studies to analyze the effectiveness, efficiency and scalability of CoDA. For example, we found that with just a little to no router pretraining, existing dense pretrained models such as T5 [Raffel et al., 2020] can be efficiently converted into CoDA models to gain both parameter efficiency and speed advantages.

## 2 Related Work

**Parameter-efficient transfer learning methods** Due to the ever-growing number of parameters in the pretrained Transformer models, various methods have been proposed for transfer learning with minimal parameter updates. Prompt tuning [Lester et al., 2021] and prefix tuning [Li and Liang, 2021] introduce new virtual token embeddings that can be finetuned as model parameters. Adapter approaches [Houlsby et al., 2019, He et al., 2021] add a small number of new, learnable parameters to each layer while keeping the pretrained parameters fixed. Another popular method, Low-Rank Adaptation [LoRA; Hu et al., 2021], injects learnable low-rank decomposition matrices into pretrained model parameters. In addition to requiring less storage cost, parameter-efficient methods have been shown to be more sample-efficient and achieve better out-of-domain generalization than standard finetuning. CoDA is an adapter approach but can be easily combined with other parameter-efficient methods such as LoRA to accelerate their inference.

**Conditional computation** The development of sparsely and conditionally activated models has been a very active research area. For example, Mixture-of-Experts (MoE) models [Shazeer et al.,

2017] and many recent advances [Du et al., 2022, Fedus et al., 2021] have been proposed to scale up the size of language models without increasing the computation cost. Many recent works have explored better token routing methods for MoE models, for example using random hashing [Roller et al., 2021], balanced assignment [Lewis et al., 2021] and expert-choosing router [Zhou et al., 2022]. CODA applies conditional computation to both attention and feed-forward blocks of the model, whereas MoE models only focus on sparse activation in the feed-forward blocks.

Similar to our approach, various recent methods have achieved computation efficiency by skipping computation on a subset of input tokens. However, the selection mechanism can be very different, such as using pooling [Nawrot et al., 2022], token merging [Bolya et al., 2023], token pruning [Rao et al., 2021, Yin et al., 2022], learned sigmoid gates [Bapna et al., 2020] and early exiting [Schuster et al., 2022]. While most of the token merging and pruning methods have been proposed for vision tasks, we show that CODA is applicable to multiple domains including text, vision and speech. In addition, token merging and our token selection method are built with different inductive biases and intuition. Token merging leverages redundancies in visual tokens, while token selection assumes a spike of token relevance. That is, only a few tokens are necessary for the prediction task. Another major difference is that CODA dynamically routes and updates token representations in each layer, whereas if a token is pruned (or merged), it will never be re-used by subsequent layers. We believe our token routing mechanism is more suited for text and speech applications, such as question answering, where different tokens might play important roles in different layers, or given different input queries.

Finally, CODA is closely related to a concurrent work, CoLT5 [Ainslie et al., 2023], which also utilizes conditional activation (token selection) for inference efficiency. The focus of CoLT5 and CODA are very different. CoLT5 specifically tailors its model architecture for long text (e.g. over 16k tokens), for example, by combining local attention with routed attention. The CoLT5 models are pre-trained from scratch and all parameters are finetuned for downstream tasks. In comparison, CODA is directly initialized and adapted from an already pretrained dense model, and we optimize its performance on parameter-efficient transfer learning. The strengths of CODA and CoLT5 can be combined for long text applications.

**Efficient Transformer models**   Many efficient Transformer variants have been proposed to accelerate model computation. Examples include creating fast attention variants [Wang et al., 2020a, Beltagy et al., 2020, Guo et al., 2022, Hua et al., 2022], searching network architectures [Press et al., 2019, So et al., 2021, Su et al., 2021] and utilizing non-attention neural modules for efficiency [Gulati et al., 2020, Lei, 2021]. CODA utilizes conditional computation as an orthogonal approach for efficiency.

**Model compression**   Apart from building efficient model architectures, model compression methods such as pruning [Han et al., 2016, Zhu and Gupta, 2017, Wang et al., 2020b, Xia et al., 2022] and distillation [Hinton et al., 2015, Kim and Rush, 2016, Turc et al., 2019, Lin et al., 2020] can be adopted to speed up model inference. Compared to these methods, CODA retains all model parameters of the pretrained large model, and therefore avoids retraining a new model from scratch or knowledge forgetting caused by parameter removal. In addition, CODA can be seen as a dynamic version of layer pruning because it can activate different Transformer layers for each token, and can be further combined with distillation to reduce the loss of accuracy caused by conditional computation.

## 3   Method

### 3.1   Architecture

Throughout this and the experiment section, we build CODA on top of parallel adapters [He et al., 2021]. However, note that our method can be generalized to other types of adapters such as sequential adapters [Houlsby et al., 2019] and LoRA [Hu et al., 2021]. We present additional experimental results using LoRA in Appendix B.3. Figure 2 illustrates our architecture and shows how CODA computes its output by selecting only a small subset of input tokens to query against the pretrained model. When parallel adapters are used, CODA introduces a small number of learnable parameters in the parallel branches, while the vast majority of model parameters (associated with the pretrained Transformer layers) remain fixed. In addition, CODA only sends $k = \lceil n/r \rceil$ tokens for heavy processing. We define $r > 1$ as the reduction factor, a constant (such as 4) to control the computation saving.

Next, we briefly introduce our notations and describe the computation of CoDA in detail. We use $F()$ to denote a parameterized neural network and the corresponding function defined by the network. For instance, a Transformer layer [Vaswani et al., 2017] consists of an attention sub-layer $F_{att}()$ followed by a feed forward sub-layer $F_{ffn}()$. Each layer also employs layer normalization [Ba et al., 2016], namely $LN_{att}()$ and $LN_{ffn}()$, before applying the attention and feed forward functions. We define $\boldsymbol{X} \in \mathbb{R}^{n \times d}$ as the input of a Transformer encoder layer, where $n$ is the number of input tokens and $d$ is the hidden size of the model.

Given layer input $\boldsymbol{X}$, we first apply layer normalization, namely $\boldsymbol{X}_{norm} = LN_{att}(\boldsymbol{X})$. The normalized input will be processed by the adapter branch and the conditional Transformer branch. Their outputs are then added and combined as the final output of the layer.

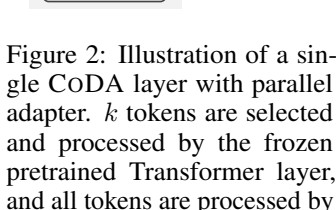

**Adapter branch**    Let $F_{adapter}()$ denote the transformation function of the adapter branch. The output is defined as

$$\boldsymbol{Z}_{adapter} = F_{adapter}(\boldsymbol{X}_{norm}) \tag{1}$$

Similar to the previous approaches, $F_{adapter}()$ is realized using a feed forward network with a small hidden size such as 64. As a result, computing $\boldsymbol{Z}_{adapter}$ only incurs a small number of floating point operations and its cost is often negligible compared to the cost of the heavy Transformer branch. The adapter branch does not conditionally select tokens. In other words, $F_{adapter}()$ is applied to all input tokens $\boldsymbol{X} \in \mathbb{R}^{n \times d}$.

Figure 2: Illustration of a single CoDA layer with parallel adapter. $k$ tokens are selected and processed by the frozen pretrained Transformer layer, and all tokens are processed by the fast adapter layer.

**Conditional branch**    The computation of the conditional branch takes three steps. First, each CoDA layer defines a router function $F_{router}()$ to select $k$ tokens for the conditional branch. The router function in each layer returns two outputs

$$\boldsymbol{m}, \boldsymbol{P} = F_{router}(\boldsymbol{X}_{norm}) \tag{2}$$

where $\boldsymbol{P} \in \{0, 1\}^{k \times n}$ is a matrix consisting of $k$ one-hot vectors indicating the selection of tokens. Here $\boldsymbol{P}[i, j] = 1$ if and only if the $i$-th selected token is the $j$-th input token from $\tilde{\boldsymbol{X}}$. $\boldsymbol{m} \in [0, 1]^n$ is a weight mask in which $\boldsymbol{m}[j]$ is the selection weight for the $j$-th input token. $\boldsymbol{m}[j] = 0$ if the token is not selected. We will describe how the router learns the selection in more details later in this section.

After the routing decision is made, the input representations of the selected tokens can be collected using a matrix multiplication,

$$\boldsymbol{X}_{routed} = \boldsymbol{P}\boldsymbol{X}_{norm} \quad \in \mathbb{R}^{k \times d} \tag{3}$$

where $k$ rows in $\boldsymbol{X}_{norm}$ are selected to construct the $k$-by-$d$ matrix $\boldsymbol{X}_{routed}$. Similar to a standard Transformer layer, the conditional branch applies attention and feed forward transformations to the selected input:

$$\bar{\boldsymbol{Z}}_{routed} = F_{att}(\boldsymbol{X}_{routed}) \tag{4}$$

$$\boldsymbol{Z}_{routed} = F_{ffn}(LN_{ffn}(\boldsymbol{X}_{routed} + \bar{\boldsymbol{Z}}_{routed})) \tag{5}$$

where $\bar{\boldsymbol{Z}}_{routed}, \boldsymbol{Z}_{routed} \in \mathbb{R}^{k \times d}$ denote the output of the attention network and the feed forward network respectively.

We consider two attention variants which differ in how they compute key-value vectors. One variant applies a *$k$-to-$k$ attention* using $\boldsymbol{X}_{routed}$ as both the query vectors and key-value vectors. The other variant applies a *$k$-to-all attention* using the entire input vectors $\boldsymbol{X}_{norm}$ as the attention keys and values. The $k$-to-all variant runs slower but obtains higher quality close to the full model. We compare the performance of the two variants in Section 5.

The attention and feed-forward output $\bar{\boldsymbol{Z}}_{routed}$ and $\boldsymbol{Z}_{routed}$ are combined and projected back to the same shape of the original input

$$\boldsymbol{Z}_{cond} = \boldsymbol{P}^{\top}(\bar{\boldsymbol{Z}}_{routed} + \boldsymbol{Z}_{routed}) \quad \in \mathbb{R}^{n \times d} \tag{6}$$

Finally $\boldsymbol{Z}_{\text{cond}}$ merges with the adapter output and the original input of the current layer to produce the output of the layer:

$$\boldsymbol{Y} = \boldsymbol{X} + \boldsymbol{Z}_{\text{adapter}} + \boldsymbol{m} \odot \boldsymbol{Z}_{\text{cond}} \tag{7}$$

$\boldsymbol{m} \odot \boldsymbol{Z}_{\text{cond}}$ is an element-wise multiplication that scales the rows of $\boldsymbol{Z}_{\text{cond}}$ using weight $\boldsymbol{m}$. This operation can be seen as a gating operation, where the hidden state $\boldsymbol{Z}_{\text{cond}}[i]$ of the $i$-th token is weighted by the token selection score $\boldsymbol{m}[i]$ assigned by the router. This enables gradient propagation from $\boldsymbol{m}$ to the router parameters, such that the token selection can be jointly optimized with other model components during training.

**Learned router**   An important ingredient of CoDA is the router function $F_{\text{router}}()$ that is learned to select a subset of tokens for favorable model performance. Given the token representation $\boldsymbol{X}_{\text{norm}}$, our router first computes dot-product score $\boldsymbol{s} = \boldsymbol{w}\,\boldsymbol{X}_{\text{norm}}^{\top}$, where $\boldsymbol{w} \in \mathbb{R}^d$ is a parameter vector associated with the router in this layer. The dot-product score $\boldsymbol{s}$ is further normalized by a function $f() : \mathbb{R}^n \to [0,1]^n$, and clipped to produce the selection score $\boldsymbol{m}$:

$$\boldsymbol{\lambda} = f(\boldsymbol{s}) \tag{8}$$
$$\boldsymbol{m} = \boldsymbol{\lambda} \odot \text{Top}(\boldsymbol{\lambda}, k) \quad \in \mathbb{R}^n \tag{9}$$

Here $\text{Top}(\boldsymbol{\lambda}, k) \in \{0,1\}^n$ is an indicator function which returns a binary mask indicating the top-$k$ values in $\boldsymbol{\lambda}$. The one-hot matrix $\boldsymbol{P}$ defined in (2) can be created according to $\text{Top}(\boldsymbol{\lambda}, k)$. In short, the highest values of $\boldsymbol{\lambda}$ will be selected by the router.

Function $f()$ must remain differentiable with respect to its input ($\boldsymbol{s}$ in this case) such that we can update the router parameters $\boldsymbol{w}$ during training. One possible choice for $f()$ is the sigmoid activation function which normalizes the values in $\boldsymbol{s}$ independently. However, this does not explicitly model the constraint that we need to select $k$ tokens from $n$ available tokens. Consider a simple case where $k = 1$, a natural choice for $f()$ would be the softmax function. Since softmax provides global normalization over the input scores, a gradient update to increase one of the scores would also decrease the other scores, a desirable effect for learning top-1 selection.

We hypothesize that a *soft top-$k$* operator that generalizes softmax should be used for general $k > 1$. This is indeed possible by formalizing soft top-$k$ as the following optimization problem:

$$f(\boldsymbol{s}) := \arg\max_{\boldsymbol{\lambda}} \; \boldsymbol{s}^{\top}\boldsymbol{\lambda} + \epsilon H(\boldsymbol{\lambda})$$
$$\text{s.t.} \quad \mathbf{1}^{\top}\boldsymbol{\lambda} = k, \;\; \boldsymbol{\lambda}[i] \in [0,1] \;\; \forall i = 1, \ldots, n \tag{10}$$

Here $H(\boldsymbol{\lambda}) = \sum_{i=1}^{n} -\boldsymbol{\lambda}[i] \log \boldsymbol{\lambda}[i]$ is a generalized entropy function (applied to any positive vector $\boldsymbol{\lambda}$ instead of a distribution), and $\epsilon > 0$ is a small coefficient.

This optimization problem is closely related to the softmax and top-$k$ operation. Specifically, when $\epsilon = 0$, it becomes a linear program which returns $\text{Top}(\boldsymbol{s}, k)$ as the solution. In addition, when $k = 1$, it can be shown that its solution is $\text{softmax}(\boldsymbol{s}/\epsilon)$. Broadly speaking, (10) will return a soft top-$k$ mask and the smoothness is controlled by $\epsilon$ (and hence $\epsilon$ must be positive to act as a temperature).

Problem (10) does not have a closed-form solution for an arbitrary $\epsilon > 0$ and $k > 1$, but its solution can be obtained using an iterative algorithm. Specifically, let $a \in \mathbb{R}$ and $\boldsymbol{b} \in \mathbb{R}^n$ be two auxiliary variables (which can be initialized to zeros). The solution takes the form $\boldsymbol{\lambda} = \exp(\frac{\boldsymbol{s}+\boldsymbol{b}+a}{\epsilon})$. The values of $a$ and $\boldsymbol{b}$ can be obtained using the following iterative updates:

$$a' = \epsilon \ln(k) - \epsilon \ln \left( \sum_{i=1}^{n} \exp \left( \frac{\boldsymbol{s}[i] + \boldsymbol{b}[i]}{\epsilon} \right) \right)$$
$$\boldsymbol{b}' = \min(-\boldsymbol{s} - a', 0) \tag{11}$$

In practice, we use $T = 20$ iterations and the function $f(\boldsymbol{s})$ returns $\exp(\frac{\boldsymbol{s}+\boldsymbol{b}+a}{\epsilon})$ using $a$ and $\boldsymbol{b}$ from the last iteration. The function $f(\boldsymbol{s})$ remain differentiable with respect to $\boldsymbol{s}$ using these iterative updates, so we can train the router jointly with other model parameters. We provide additional discussion and the derivation of the updates in Appendix §C.

| Model | Reduction $r$ | Base | | | Large | | | XL | | | $\Delta$ Avg |
|---|---|---|---|---|---|---|---|---|---|---|---|
| | | MNLI | RTE | BoolQ | MNLI | RTE | BoolQ | MNLI | RTE | BoolQ | |
| Parallel Adapter (w/o conditional computation) | - | 87.1 | 71.5 | 77.9 | 90.3 | 84.8 | 85.8 | 91.5 | 89.9 | 88.4 | ±0.0 |
| **CoDA, $k$-to-all attention** | 3 | 86.6 | 72.6 | 76.6 | 90.2 | 85.9 | 85.1 | 91.4 | 91.3 | 89.4 | +0.2 |
| CoDA, $k$-to-$k$ attention | | 86.3 | 72.2 | 76.2 | 89.8 | 87.0 | 83.7 | 91.4 | 89.5 | 88.2 | −0.3 |
| **CoDA, $k$-to-all attention** | 5 | 86.0 | 70.8 | 76.0 | 89.7 | 85.2 | 84.3 | 91.0 | 91.3 | 87.2 | −0.6 |
| CoDA, $k$-to-$k$ attention | | 82.5 | 70.8 | 75.4 | 88.1 | 87.0 | 81.8 | 89.9 | 87.7 | 84.8 | −2.1 |

Table 2: Results of applying CoDA to T5 v1.1 models. CoDA achieves significant computation savings while retaining accuracy close to the dense baseline. We compare CoDA to a corresponding parallel adapter method that processes all tokens without conditional computation. We report accuracy on the development set on 3 tasks × 3 model sizes, and set the number of selected tokens $k = \lceil n/r \rceil$. The last column shows the change on average accuracy with respect to the parallel adapter method. We select the $k$-to-all version as our default (shown in bold).

## 3.2 Training

CoDA can be directly initialized from an existing Transformer model. Given a pretrained model such as T5 [Raffel et al., 2020], the Transformer layers are directly re-used and copied in the conditional branches of CoDA, and only the adapter and router parameters are randomly initialized. Because pretraining a large dense model can be expensive, our method reduces the overall training cost.

The routers and neural network components in CoDA must co-operate and be optimized for accurate model predictions. When the available finetuning data is limited, a random initialization for the router (and adapter) parameters can be sub-optimal. We demonstrate that CoDA can be further pretrained using the same pretraining objective as the dense model, in order to enhance downstream performance. Importantly, CoDA requires significantly fewer training steps during pretraining, since most of its parameters are taken from an already pretrained model. We show that the cost of CoDA pretraining can be 10-30x lower than the pretraining of its original dense model. We present this analysis in Section 5.

Finally, we train CoDA on downstream tasks by only updating the adapter, router and layer normalization parameters. The size of the adapters is small (e.g. 5M parameters), and each router and layer normalization block only introduces $d$ parameters, where $d$ is the model dimension. As a result, CoDA remains parameter-efficient similar to previous adapter and prompt-tuning methods.

## 4 Experimental setup

CoDA is evaluated on three domains including natural language processing (NLP), computer vision and speech processing, and on a range of applications such as classification, question answering, summarization and speech recognition. The experiments are organized as follows: We first demonstrate the effectivenss of CoDA conduct analyses on its design choices using the publicly available T5 models (§5). In our final results (§6), we pretrain Transformer models from scratch and extend our evaluation to vision and speech domains.

**Datasets** We use the C4 corpus [Raffel et al., 2020] for pretraining text models. For speech models, we use the LibriLight corpus [Kahn et al., 2020] for pretraining. Our vision Transformer models use the same data and training procedure in Pix2Struct [Lee et al., 2022]. Our finetuning datasets for text models include the MNLI [Williams et al., 2018], RTE [Dagan et al., 2005, Haim et al., 2006, Giampiccolo et al., 2007, Bentivogli et al., 2009], BoolQ [Clark et al., 2019], SQuAD [Rajpurkar et al., 2016] and XSum [Narayan et al., 2018] datasets. The speech models are evaluated on the speech recognition task using the LibriSpeech dataset [Panayotov et al., 2015]. Finally, we use the OCR-VQA [Mishra et al., 2019], DocVQA [Mathew et al., 2021], and Screen2Words [Wang et al., 2021] datasets for vision models.

|  | | Base | | | Large | | | XL | | | |
| Model | Reduction $r$ | MNLI | RTE | BoolQ | MNLI | RTE | BoolQ | MNLI | RTE | BoolQ | $\Delta$ Avg |
|---|---|---|---|---|---|---|---|---|---|---|---|
| **Soft top-$k$** | | 86.3 | 72.2 | 76.2 | 89.8 | 87.0 | 83.7 | 91.4 | 89.5 | 88.2 | $\pm0.0$ |
| Sigmoid gate as $f(s)$ | 3 | 85.7 | 70.8 | 72.8 | 89.2 | 82.3 | 81.0 | 90.6 | 88.1 | 86.2 | $-2.0$ |
| Truncation – selecting first $k$ | | 81.1 | 70.8 | 72.7 | 84.9 | 77.3 | 82.3 | 85.6 | 84.5 | 85.4 | $-4.4$ |
| **Soft top-$k$** | | 82.5 | 70.8 | 75.4 | 88.1 | 87.0 | 81.8 | 89.9 | 87.7 | 84.8 | $\pm0.0$ |
| Sigmoid gate as $f(s)$ | 5 | 82.9 | 71.5 | 72.1 | 86.7 | 82.3 | 80.1 | 88.3 | 87.0 | 82.4 | $-1.6$ |
| Truncation – selecting first $k$ | | 62.2 | 64.6 | 71.1 | 64.9 | 70.4 | 75.4 | 66.6 | 76.2 | 81.1 | $-12.9$ |

Table 3: Ablation study on routing methods. We use CODA $k$-to-$k$ variant for a fair comparison with the truncation method. Better routing method delivers better accuracy on various tasks and model sizes tested. We use soft top-$k$ as our default method.

## 5   Understanding and Analyzing CODA

**Setup**   We present several analyses to validate the design choices of CODA in this section. We initialize CODA using the version 1.1 release of T5 checkpoints[2], and perform CODA pretraining using the same setting as the T5 models. During pretraining, we set routing capacity to $k = 192$ given input sequence length $n = 512$. We do not tune the value of $k$ for pretraining, but will report the results of using different $k$ values in finetuning. We perform 100K gradient steps, which is 10% of the total number of steps used to train the T5 dense models. The overall computational cost is over 20x less than the full training of dense models, since CODA only applies heavy computation on less than half of the tokens.

For simplicity, we evaluate on classification tasks for various ablation studies of CODA. Specifically, we report results on the MNLI, RTE and BoolQ datasets, and test three different model sizes including the Base, Large and XL size of T5. We will extend our evaluation to generation tasks such as question answering in the full result section (§6).

**Can CODA be fast and accurate?**   Table 2 presents the finetuning results of CODA. As a comparison, we also report the results of Parallel Adapter, which is similar to CODA except that it applies the expensive Transformer layers to all input tokens. This constitutes an upper-bound, and is a strong baseline that has been reported as the best among a range of adapter and prompt tuning methods [He et al., 2021]. As shown in Table 2, CODA can achieve 3-5x computation reduction ($r = 3, 5$) in the Transformer layers at a cost of less than 1.0 point drop on average accuracy. As expected, our $k$-to-all attention variant achieves consistently better accuracy than the $k$-to-$k$ variant, since it can access the full attention context. On the other hand, the $k$-to-$k$ attention variant runs faster in practice, which can be beneficial for tasks with very long inputs. We select the $k$-to-all version in the final result section (§6).

**How many pretraining steps are needed?**   Figure 3 plots the finetuning accuracy by varying the number of pretraining steps for CODA. Because CODA can be initialized using pretrained dense models, it requires as few as 20K steps to obtain competitive finetuning results. Of course, using more pretraining steps can improve the downstream accuracy. The fact that CODA can be quickly updated without repeating the expensive pretraining will be very beneficial in real-world applications.

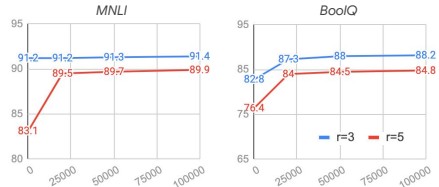

Figure 3: Finetuning accuracy (y-axis) as a function of CODA pretraining steps (x-axis). We show results using 0, 20K, 50K and 100K pretraining steps, and for reduction factor $r = 3$ and $r = 5$ respectively. CODA requires as few as 20K steps to obtain competitive finetuning accuracy.

**Does learned routing matter?**   We analyze the impact of learned routing in Table 3 by comparing our soft top-$k$ router with other router implementations. We implement a variant that replaces soft top-$k$ with the sigmoid activation function, so the selection weight of each token activates on its own (without considering the capacity constraint). As shown in the table, this variant

---

[2]`https://github.com/google-research/text-to-text-transfer-transformer/blob/main/released_checkpoints.md#t511`

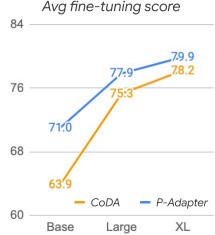

| Model | Trainable Params | Reduction $r$ | MNLI Acc. | RTE Acc. | BoolQ Acc. | SQuAD F1 | ReCord F1 | XSum R2 | ΔAvg |
|---|---|---|---|---|---|---|---|---|---|
| Parallel Adapter | 10M | - | 91.5 | 91.0 | 88.5 | 94.8 | 91.4 | 21.9 | ±0.0 |
| CoDA | 10M | 3 | 91.2 | 90.3 | 87.5 | 94.1 | 89.3 | 20.6 | −1.0 |
| CoDA | 10M | 5 | 90.7 | 89.5 | 87.3 | 93.5 | 87.6 | 20.2 | −1.7 |
| Prefix Tuning [Li and Liang, 2021]† | 15M (2M) | - | (86.3) | - | - | - | - | 20.5 | |
| Sequential Adapter [Houlsby et al., 2019]† | 10M (2M) | - | (87.2) | - | - | - | - | 20.0 | |
| Parallel Adapter [He et al., 2021]† | 10M | - | - | - | - | - | - | 20.7 | |

Table 4: Comparison of CoDA and parallel adapter on 6 language tasks. We report results on the test set of XSum, and on the development set of other tasks. † indicates results taken from He et al. [2021], and referenced results in bracket correspond to using 2M adapter parameters. Note that our Parallel Adapter numbers are stronger as our pretrained Transformer backbone uses more parameters than the model used in He et al. [2021].

Figure 4: Average finetuning scores of Parallel Adapter and CoDA at different model sizes.

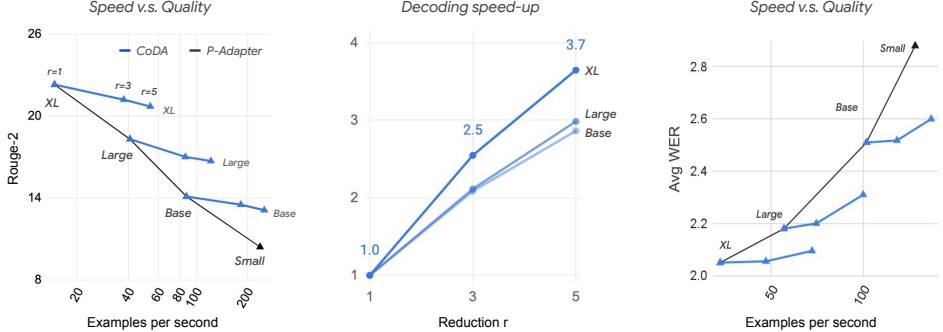

Figure 5: The scaling of CoDA on the XSum and LibriSpeech dataset. Left: CoDA achieves better speed-quality trade-off than finetuning adapters with smaller models, on the XSum dataset. Middle: larger CoDA model achieves higher speed-ups. Right: CoDA achieves better speed-quality trade-off than the dense baseline on the LibriSpeech dataset.

achieves worse accuracy on almost all tasks and model sizes tested, getting 2.0 point worse on average. We also implement a "no-learning" baseline that simply selects the first $k$ tokens, which is equivalent to truncating the input sequence.[3] This baseline performs much worse, resulting in more than 10 point decrease in accuracy for small $k$ (and equivalently large $r$). This analysis confirms the importance of learning a good routing in order to retain strong model performance.

## 6 Full Results

**Setup** In this section, we apply our best training recipe to all tasks and application domains. We first pretrain dense Transformer models, followed by the CoDA training procedure in §3.2. Our speech models are pretrained using a masked language modeling (MLM) objective similar to BERT [Devlin et al., 2019], and random quantized output label space [Chiu et al., 2022]. Our vision and text models use an encoder-decoder architecture similar to T5 but incorporate a few changes. Following PaLM [Chowdhery et al., 2022], we use multi-query attention [Shazeer, 2019] that shares the same key and value projection for multiple query heads. We only use 6 decoder layers and increase the feed forward hidden size (to compensate for the decrease in the number of layers). These modifications have a neutral effect on model quality, but speed up auto-regressive decoding significantly. We will show CoDA is compatible with these changes and can further speed up inference by a considerably large factor. We provide more details of our experimental setup in Appendix A.

**NLP results** In addition to the classification datasets used in Section 5, we also evaluate our final models on the SQuAD, ReCord and XSum datasets which require generating an answer or a summary

---

[3]We always include the question text for BoolQ, to achieve higher accuracy.

| Model | r | Base | | Large | | XL | |
|---|---|---|---|---|---|---|---|
| | | clean | other | clean | other | clean | other |
| w2v-BERT | - | 1.8 | 3.6 | 1.5 | 2.9 | 1.5 | 2.9 |
| BEST-RQ | - | 1.7 | 3.5 | 1.6 | 2.9 | 1.4 | 2.7 |
| P-Adapter | - | 1.6 | 3.5 | 1.4 | 3.0 | 1.4 | 2.7 |
| CODA | 2 | 1.6 | 3.5 | 1.4 | 3.0 | 1.4 | 2.8 |
| CODA | 4 | 1.6 | 3.6 | 1.5 | 3.1 | 1.4 | 2.8 |

Table 5: Comparison of CODA and the parallel adapter baselines on Librispeech. We report the WER results on test-clean and test-other. More results can be found in §B.2.

| Model | r | OCRVQA | | DocVQA | | Screen2Words | |
|---|---|---|---|---|---|---|---|
| | | EM | Speedup | ANLS | Speedup | CIDEr | Speedup |
| Parallel Adapter | - | 67.5 | 1× | 70.8 | 1× | 110.2 | 1× |
| CODA | 4 | 68.2 | 4.6× | 71.8 | 4.6× | 111.6 | 4.6× |
| CODA | 8 | 67.6 | 8.0× | 66.6 | 8.0× | 108.1 | 8.0× |
| CODA | 16 | 66.9 | 13.5× | 56.6 | 12.1× | 109.0 | 12.5× |
| CODA | 32 | 64.4 | 19.4× | 42.5 | 16.7× | 104.2 | 17.8× |

Table 6: Comparison of CODA and the parallel adapter applied to a pretrained Pix2Struct model [Lee et al., 2022] on 3 visually-situated language understanding tasks.

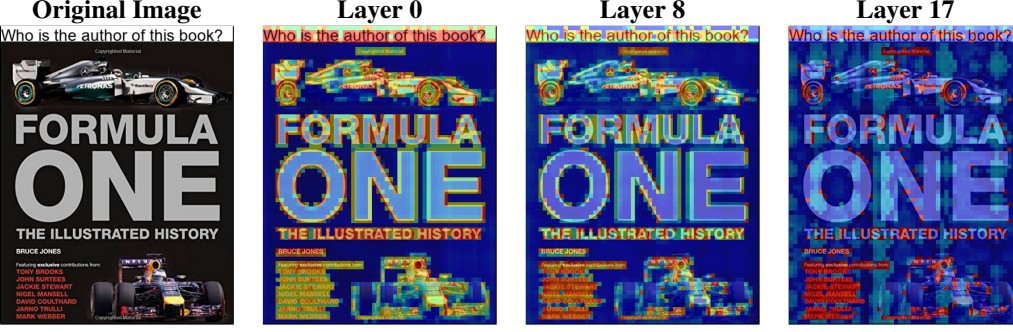

Figure 6: Visualization of routing preferences for a CODA model applied to the OCR-VQA task. Warmer and cooler colors represent higher and lower scores respectively. Router prefers diverse coverage in early layers, but converges to selecting sparse and representative patches in later layers.

given the input. Table 4 contains the finetuning results of XL models. Compared to the parallel adapter baseline that uses full computation, CODA achieves 3x and 5x computation reduction with only 1.0 and 1.7 point loss in average score.

Figure 4 and 5 highlight the scaling trend of CODA. CODA runs much faster with slightly worse quality than the parallel adapter baseline. This is expected because the baseline processes all tokens in every layer, whereas CODA only selects $1/r$ of tokens for heavy processing. Importantly, this quality gap reduces as the model size increases (as shown in Figure 4), making CODA a computationally efficient choice for large models. Indeed, CODA can trade off quality for speed by varying the number of selected tokens. Figure 5 (left) demonstrates that CODA achieves much stronger speed-quality trade-off compared to dense models without conditional computation. The black line indicates the results of Parallel Adapter when the model size grows from Small to XL, and each blue line represents the speed-quality trade-off of CODA using $r = 1, 3, 5$. Moreover, Figure 5 (middle) shows that larger CODA models exhibit higher inference speed-ups. These observations are consistent on other tasks. We provide additional results in Appendix §B.

**Speech recognition results** We further validate the performance of CODA in the speech domain. Our model uses a Transformer encoder and a 2-layer LSTM Transducer [Graves, 2012]. Similar to NLP setups, we test the performance of the speech model on 3 scales – Base, Large and XL (see Appendix A for details). Table 5 demonstrates that with sizable reduction ratios ($r = 2, 4$), the change on word error rate (WER) is consistently minimal on the test-clean and test-other sets of LibriSpeech across different model sizes (and on other sets in §B.2). Moreover, our results are comparable to the top-performing models, such as w2v-BERT [Chung et al., 2021] and BEST-RQ [Chiu et al., 2022], that are fully finetuned by updating all parameters. Figure 5 (right) highlight again that applying conditional computation leads to better speed-quality trade-off compared to dense models.

**Vision results** We extend our experiments to visual tasks that involves natural language within the image, such as documents and user interfaces. Our experiments are based on Pix2Struct [Lee et al., 2022], where an image-encoder-text-decoder is pretrained by learning to predict simplified HTML from webpage screenshots. Table 6 shows the results on three tasks that were also evaluated in the

original Pix2Struct paper. In OCRVQA and Screen2Words, we observe relatively small drops in performance when reducing the number of routed tokens (i.e. patches). When the capacity is 1/16th of the original sequence length, leading to around $13\times$ speedup, we only lose about 1 point. We speculate that this is due to the high-level sparsity in the inputs for these two tasks. For DocVQA, where there is comparatively more textual information, we observe a steeper performance-speed trade-off but still achieve a $8\times$ speedup with a 4-point drop.

To provide a more intuitive understanding why CODA works, we visualize the router behavior for the OCR-VQA model in Figure 6. We show which patches the routers prefers the most (warmest colors) and least (coolest colors), for several layers. The first, immediately obvious, observation is that router avoids low-frequency patches, i.e. patches likely to be "whitespace", since they can be adequately handled by the cheap adapter layers. The second, more subtle, observation is that the router progressively converges on a small number of key patches that we hypothesize serve as representations for larger regions. The visualization confirms that CODA is able to select meaningful and representative patches that are useful for the prediction task.

# 7 Conclusion and Limitation

We present CODA, a parameter-efficient adapter method that enables fast inference. CODA relies on conditional computation to selectively activate model computation on important input units, providing a novel way to balance model expressivity and efficiency.

In this work, we focus on encoder-heavy applications such as summarization, speech recognition and visual question answering, by applying our method to the encoder. One limitation of CODA is that the current routing mechanism (i.e. token selection in a given sequence) is not directly applicable to decoder-only models for auto-regressive token generation. Enabling fast token generation using conditional activation in decoder layers is an interesting direction we plan to explore in future work.

# 8 Acknowledgements

We would like to thank Rama Pasumarthi, Hongkun Yu, Kelvin Guu, Zhuyun Dai, Timothy Dozat, Raphael Hoffmann, Tao Wang, Tal Schuster, Ziwei Ji, Frederick Liu and Slav Petrov for helpful advice and discussion.

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

# A  Experimental details

**Model implementation**  For our text and vision experiments, we implement our models using JAX [Bradbury et al., 2018]. Specifically, our training and model modules are built on top of the T5X, Flax and Flaxformer framework [Roberts et al., 2022, Heek et al., 2020]. Following the T5 v1.1 implementation and PaLM [Chowdhery et al., 2022], our Transformer models use the GLU variant [Shazeer, 2020] as the feed forward network and multi-query-attention [Shazeer, 2019] as the attention block. These modifications are shown to enhance modeling capacity and speed up decoding respectively.

For the speech experiments, we use TensorFlow [Abadi et al., 2015] and the Lingvo framework [Shen et al., 2019]. The state-of-the-art Transformer variant for speech recognition is the Conformer architecture [Gulati et al., 2020] which additionally uses depth-wise convolution in each layer. Since the convolution operation is applied to consecutive inputs and does not immediately support routing, we use the standard Transformer architecture [Vaswani et al., 2017] instead. Swish activation is used in the feed forward blocks, following Gulati et al. [2020]. We provide the model configuration details in Table 7.

| Model | Num of params | Layers | Num of heads | $d_{\text{model}}$ | $d_{\text{ffn}}$ | $d_{\text{head}}$ | $d_{\text{adpt}}$ |
|---|---|---|---|---|---|---|---|
| Text Base | 0.1B | 12, 6 | 12 | 768 | 3072 | 128 | 64 |
| Text Large | 0.5B | 24, 6 | 16 | 1024 | 4096 | 128 | 64 |
| Text XL | 2.1B | 24, 6 | 32 | 2048 | 8192 | 128 | 64 |
| Speech Base | 0.2B | 31, 2 | 8 | 768 | 3072 | 96 | 256 |
| Speech Large | 0.6B | 32, 2 | 8 | 1280 | 5120 | 160 | 256 |
| Speech XL | 1.1B | 32, 2 | 8 | 1664 | 6656 | 208 | 256 |
| Vision | 0.7B | 18, 6 | 24 | 1536 | 3968 | 128 | 256 |

Table 7: Configuration of Transformer models used in §6. We show the total number of parameters (in billions). $d_{\text{model}}$ is the model hidden dimension, $d_{\text{ffn}}$ is the intermediate FFN hidden dimension, $d_{\text{head}}$ is the attention head dimension and $d_{\text{adpt}}$ is the adapter hidden dimension.

| Dataset | Input length | Batch size | Steps | Optimizer | Learning rate |
|---|---|---|---|---|---|
| MNLI | 128 | | | | |
| RTE | 256 | | | | |
| BoolQ | 384 | 128 | 300K | Adafactor | 0.001, constant |
| SQuAD | 512 | | | | |
| ReCord | 512 | | | | |
| XSum | 1664 | | | | |
| LibriSpeech | 3200 | 256 | 150K | Adafactor | 0.001, inverse decay |
| OCR-VQA | 4096 | 256 | | | |
| DocVQA | 4096 | 256 | 20K | Adafactor | 0.01, cosine decay |
| Screen2Words | 4096 | 32 | | | |

Table 8: Fine-tuning hyperparmaeters. We use a maximum of 300K steps for NLP tasks following T5 [Raffel et al., 2020]. Pix2struct [Lee et al., 2022] uses 10K fine-tuning steps for vision tasks. We use 20K steps as CODA takes longer to train.

**Model training**  We use the same data and procedure described in T5 [Raffel et al., 2020], BEST-RQ [Chiu et al., 2022] and Pix2struct [Lee et al., 2022] for pre-training the respective text, speech and vision models. We use the same training hyper-parameters, such as batch size, input sequence length, the number of pre-training steps and the choice of optimizer and learning rate scheduling. All models have been pre-trained using 128 or 256 TPUv3/TPUv4 chips.

We run CODA pre-training for text and vision models, using an additional 100K steps and 200K steps respectively. For text models, the input sequence length is $n = 512$ and we set the number of selected tokens $k = 192$. For vision models, the input sequence contains $n = 4096$ image patches and we set $k = 1024$. CODA pre-training is not used for our speech models because there are

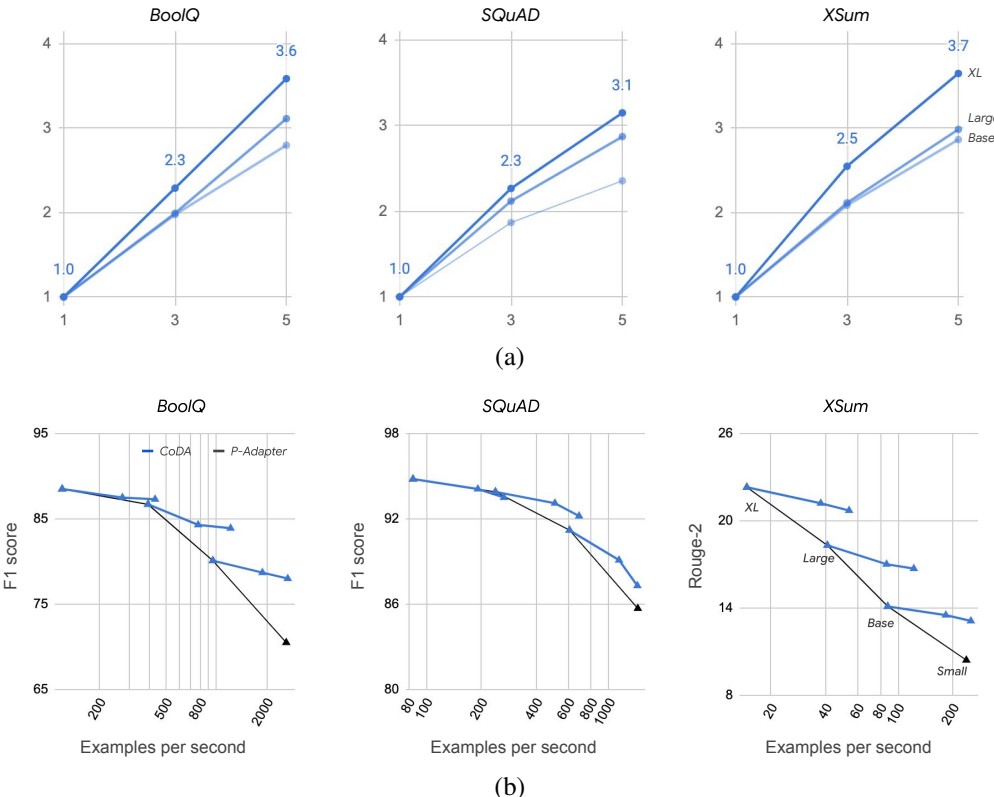

Figure 7: Analyzing the speed and quality of CODA. We select 3 representative tasks including BoolQ (classification), SQuAD (question answering) and XSum (summarization). (a) Relative decoding speed given different reduction factor $r$. (b) Speed-quality trade-off of CODA applied on different tasks and model sizes. CODA achieves better quality than the dense Parallel Adapter baseline when running at a similar inference speed. The black line shows the performance of Parallel Adapter given Small to XL model size. Each blue line represents the performance of CODA for reduction $r = 1, 3, 5$ for a given model size. When $r = 1$, CODA is equivalent to the dense baseline.

sufficient fine-tuning data. Following standard practice in speech, we use the 1K hour data from the LibriSpeech dataset [Panayotov et al., 2015] and another 30K hour data generated using the noisy student self-training method [Xie et al., 2020, Zhang et al., 2022].

Table 8 lists the hyper-parameters used for fine-tuning, including the sequence length, learning rate, batch size and the number of fine-tuning steps used. For NLP datasets, we set the maximum input length and decoding length to the 98th percentile of lengths in the training set. For vision datasets, we set the input length following the suggested values in Pix2struct. We also find that annealing the number of routed tokens $k$ can achieve better finetuning results. Specifically, we decrease $k$ linearly from the sequence length $n$ down to the target value $n/r$ using the first 10% to 20% of the finetuning steps.

# B  Additional results

## B.1  NLP

Table 9 contains the complete fine-tuning results on the 6 language datasets. As discussed in §6, the gap between CODA and its counterpart without conditional computation is large at Base size. As the model size increases, CODA retains almost the same level of quality given 3x computation reduction ($r = 3$). The reduction leads to decoding speed-ups, as shown in Figure 7. More importantly, we see that larger model benefits more from CODA, achieving a speed-up factor close to the reduction factor

| Model | Trainable Params | Reduction $r$ | MNLI Acc. | RTE Acc. | BoolQ Acc. | SQuAD F1 | ReCord F1 | XSum R2 |
|---|---|---|---|---|---|---|---|---|
| | | | | | Base | | | |
| Parallel Adapter | | - | 88.2 | 75.8 | 80.1 | 91.2 | 76.7 | 14.1 |
| CoDA | 2M | 3 | 85.8 | 68.6 | 78.7 | 89.1 | 69.1 | 13.5 |
| CoDA | | 5 | 82.8 | 60.3 | 78.0 | 87.3 | 61.8 | 13.1 |
| | | | | | Large | | | |
| Parallel Adapter | | - | 90.5 | 90.6 | 86.7 | 93.9 | 87.2 | 18.3 |
| CoDA | 5M | 3 | 90.0 | 84.8 | 84.3 | 93.1 | 84.6 | 17.0 |
| CoDA | | 5 | 89.4 | 88.5 | 83.9 | 92.2 | 81.2 | 16.7 |
| | | | | | XL | | | |
| Parallel Adapter | | - | 91.5 | 91.0 | 88.5 | 94.8 | 91.4 | 22.3 |
| CoDA | 10M | 3 | 91.2 | 90.3 | 87.5 | 94.1 | 89.3 | 21.2 |
| CoDA | | 5 | 90.7 | 89.5 | 87.3 | 93.5 | 87.6 | 20.7 |

Table 9: Fine-tuning results on 6 language tasks $\times$ 3 model sizes. We report the best results on the development set.

| Model | $r$ | Base | | | | Large | | | | XL | | | | $\Delta$ Avg |
|---|---|---|---|---|---|---|---|---|---|---|---|---|---|---|
| | | dev | | test | | dev | | test | | dev | | test | | |
| | | clean | other | clean | other | clean | other | clean | other | clean | other | clean | other | |
| w2v-BERT | - | 1.7 | 3.6 | 1.8 | 3.6 | 1.5 | 2.9 | 1.5 | 2.9 | 1.5 | 2.6 | 1.5 | 2.9 | - |
| BEST-RQ | - | 1.6 | 3.5 | 1.7 | 3.5 | 1.5 | 2.8 | 1.6 | 2.9 | 1.4 | 2.7 | 1.4 | 2.7 | - |
| Parallel Adapter | - | 1.5 | 3.5 | 1.6 | 3.5 | 1.4 | 3.0 | 1.4 | 3.0 | 1.4 | 2.7 | 1.4 | 2.7 | $\pm 0.0$ |
| CoDA | 2 | 1.5 | 3.5 | 1.6 | 3.5 | 1.4 | 3.0 | 1.4 | 3.0 | 1.4 | 2.7 | 1.4 | 2.8 | $+\mathbf{0.01}$ |
| CoDA | 4 | 1.6 | 3.6 | 1.6 | 3.6 | 1.4 | 3.1 | 1.5 | 3.1 | 1.4 | 2.8 | 1.4 | 2.8 | $+\mathbf{0.07}$ |

Table 10: Comparison of CoDA and the parallel adapter baselines on all 4 splits (dev-clean, dev-other, test-clean, test-other) of Librispeech.

$r$. These results highlight the potential of CoDA for large-scale models, which we plan to investigate in future work.

## B.2 Speech

Table 10 extends Table 5 by including WER results on dev-clean and dev-other splits. From the table, one can observe that XL with CoDA ($r = 2, 4$) are consistently better than the Large parallel adapter model on each split, and the Large model with CoDA ($r = 2, 4$) are also consistently better than the Base PA on each split. Given the inference speeds for CoDA models shown in Table 8, larger CoDA models are generally faster and better than smaller dense ones (even with PA) with regard to either time cost or computation GFLOPs. Therefore, it is likely for CoDA to help scale up ASR models with decent computation resources and time cost.

## B.3 Combining CoDA and LoRA

CoDA can be easily combined with other types of adapter methods. To see this, we additionally implemented a variant that combines with Low-Rank Adapter [LoRA; Hu et al., 2021], which is another parameter-efficient transfer learning method that recently became the most popular choice for LLMs. We incorporate the latest development suggested in the QLoRA paper [Dettmers et al., 2023], which adds low-rank adapters to every linear projection matrix in the Transformer layers. This is found to obtain better fine-tuning performance than the original implementation. Our CoDA variant with LoRA simply removes the parallel adapter branches and instead adds low-rank adapters to the projection matrices of the pretrained layers.

Table 11 shows the finetuning results. The new LoRA baseline achieves stronger accuracy than the Parallel Adapter baseline (84.0 v.s. 82.9 on average), highlighting the effectiveness of recent development on LoRA. In addition, our CoDA variant using LoRA still achieves very close accuracy compared to its dense counterpart (84.0 v.s. 84.0 or 83.7 on average). We believe the additional

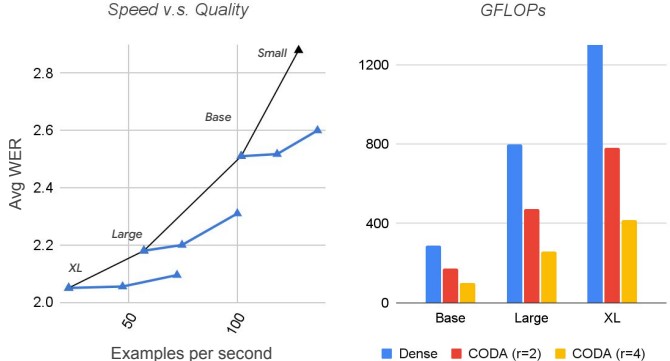

Figure 8: The scaling of CoDA on Librispeech. Left: Larger models with CoDA can achieve much better performance with the same or faster speed compared to smaller models. Right: GFLOPS reduction is significant as $r$ increases.

| Model | Reduction $r$ | Base | | | Large | | | |
| | | MNLI | RTE | BoolQ | MNLI | RTE | BoolQ | Avg on 6 |
|---|---|---|---|---|---|---|---|---|
| Parallel Adapter (PA) | - | 87.1 | 71.5 | 77.9 | 90.3 | 84.8 | 85.8 | 82.9 |
| CoDA w/ PA | 3 | 86.6 | 72.6 | 76.6 | 90.2 | 85.9 | 85.1 | 82.8 |
| CoDA w/ PA | 5 | 86.0 | 70.8 | 76.0 | 89.7 | 85.2 | 84.3 | 82.0 |
| Low-rank Adapter (LoRA) | - | 88.0 | 73.7 | 80.3 | 90.7 | 85.2 | 86.3 | 84.0 |
| CoDA w/ LoRA | 3 | 86.2 | 76.9 | 78.4 | 90.3 | 86.3 | 85.8 | 84.0 |
| CoDA w/ LoRA | 5 | 86.0 | 76.9 | 78.3 | 89.8 | 86.3 | 84.7 | 83.7 |

Table 11: Results of applying CoDA with Parallel Adapter v.s. Low-Rank Adapter. We report results on the development sets and on the Base and Large model size.

results strengthen our claims – that CoDA enables a strong trade-off between accuracy and efficiency using conditional activation, and this technique can be combined with other developments in PETL.

## C   Soft top-$k$ algorithm

### C.1   Derivation of the iterative updates

We present the derivation of iterative updates (11) for solving the soft top-$k$ problem (10) in Section 3. The soft top-$k$ operation is defined as a maximization problem (10). For the derivation, we rewrite it as an equivalent minimization problem:

$$\max_{\boldsymbol{\lambda}} \quad \boldsymbol{s}^\top \boldsymbol{\lambda} + \epsilon H(\boldsymbol{\lambda})$$
$$\Longleftrightarrow \quad \min_{\boldsymbol{\lambda}} \quad -\boldsymbol{s}^\top \boldsymbol{\lambda} - \epsilon H(\boldsymbol{\lambda})$$
$$\Longleftrightarrow \quad \min_{\boldsymbol{\lambda}} \quad -\boldsymbol{s}^\top \boldsymbol{\lambda} - \epsilon H(\boldsymbol{\lambda}) - \epsilon \mathbf{1}^\top \boldsymbol{\lambda} \tag{12}$$
$$\text{s.t. } \mathbf{1}^\top \boldsymbol{\lambda} = k, \ \ \boldsymbol{\lambda}[\boldsymbol{i}] \in [0,1], \ \ i = 1, \dots, n.$$

Note the term $\epsilon \mathbf{1}^\top \boldsymbol{\lambda}$ will be a constant $\epsilon \times k$, but we include it in the minimization object to make our derivation simpler later.

Now, let $a \in \mathbb{R}$ and $\boldsymbol{b} \in \mathbb{R}^m$ be the Lagrangian variables corresponding to the linear constraints $\mathbf{1}^\top \boldsymbol{\lambda} = k$ and $\boldsymbol{\lambda}[i] \leq 1 \ \forall i$ .[4] The minimization problem is equivalent to its Lagrangian expression:

$$\min_{\boldsymbol{\lambda} \in \mathbb{R}^m} \ \max_{a \in \mathbb{R}, \boldsymbol{b} \leq \mathbf{0}} \ -\boldsymbol{s}^\top \boldsymbol{\lambda} - \epsilon H(\boldsymbol{\lambda}) - \epsilon \mathbf{1}^\top \boldsymbol{\lambda} + a(k - \mathbf{1}^\top \boldsymbol{\lambda}) + \boldsymbol{b}^\top (\mathbf{1} - \boldsymbol{\lambda}) \tag{13}$$

---

[4] $\boldsymbol{\lambda}[i] \geq 0 \ \forall i$ is already implied by the term $H(\boldsymbol{\lambda}) = \sum_i -\boldsymbol{\lambda}[i] \log \boldsymbol{\lambda}[i]$ in the objective, due to the use of $\log \boldsymbol{\lambda}[i]$.

The objective function (12) is strongly convex and the solution space of $\boldsymbol{\lambda}$ is a convex set. As a result, strong duality holds so we can instead solve the dual problem. The dual problem exchanges the $\min$ and $\max$ operators in (13):

$$\max_{a \in \mathbb{R}, \boldsymbol{b} \leq \boldsymbol{0}} \min_{\boldsymbol{\lambda} \in \mathbb{R}^m} -\boldsymbol{s}^\top \boldsymbol{\lambda} - \epsilon H(\boldsymbol{\lambda}) - \epsilon \boldsymbol{1}^\top \boldsymbol{\lambda} + a(k - \boldsymbol{1}^\top \boldsymbol{\lambda}) + \boldsymbol{b}^\top (\boldsymbol{1} - \boldsymbol{\lambda}) \tag{14}$$

The optimal solution $(a, \boldsymbol{b}, \boldsymbol{\lambda})$ must have the Karush-Kuhn-Tucker (KKT) conditions hold [Kuhn and Tucker, 2014], namely

$$\frac{\partial \left( -\boldsymbol{s}^\top \boldsymbol{\lambda} - \epsilon H(\boldsymbol{\lambda}) + \epsilon \boldsymbol{1}^\top \boldsymbol{\lambda} + a(k - \boldsymbol{1}^\top \boldsymbol{\lambda}) + \boldsymbol{b}^\top (\boldsymbol{1} - \boldsymbol{\lambda}) \right)}{\partial \boldsymbol{\lambda}} = 0$$

$$\iff \quad \boldsymbol{\lambda} = \exp\left( \frac{\boldsymbol{s} + a + \boldsymbol{b}}{\epsilon} \right) \quad \iff \quad \boldsymbol{\lambda}[i] = \exp\left( \frac{\boldsymbol{s}[i] + a + \boldsymbol{b}[i]}{\epsilon} \right) \quad \forall i = 1, \ldots, n$$

Substituting $\boldsymbol{\lambda}$ using the above equation in (14), the dual problem now has a simple form:

$$\max_{a \in \mathbb{R}, \boldsymbol{b} \leq \boldsymbol{0}} k \cdot a + \boldsymbol{1}^\top \boldsymbol{b} - \boldsymbol{1}^\top \exp\left( \frac{\boldsymbol{s} + a + \boldsymbol{b}}{\epsilon} \right)$$

We can solve this problem using coordinate descent [Wright, 2015] by successively maximizing the function with either $a$ or $\boldsymbol{b}$ fixed. That is, we find the optimal $a$ that maximizes the dual objective given a fixed $\boldsymbol{b}$, and vice versa. This leads to the iterative updates (11) described in Section 3.

$$a' = \epsilon \ln(k) - \epsilon \ln \left( \sum_{i=1}^{n} \exp\left( \frac{\boldsymbol{s}[i] + \boldsymbol{b}[i]}{\epsilon} \right) \right), \qquad \boldsymbol{b}' = \min(-\boldsymbol{s} - a', 0)$$

In short, we obtain the iterative updates of the soft top-$k$ problem (10) by solving its dual problem and by performing coordinate decent of the dual variables $a$ and $\boldsymbol{b}$. The iterative updates are in fact the coordinate decent steps.

## C.2   The $\epsilon$-scaling trick

The iterations of $a$ and $\boldsymbol{b}$ will converge but the number of iterations needed can be very large for small $\epsilon$. In practice, we only perform a small number of iterations and return the corresponding $\boldsymbol{\lambda}$, which may be close but not the exact solution to (12). In order to improve the convergence given a small number of iterations, we apply an empirical trick called the $\epsilon$-*scaling heuristic* [Schmitzer, 2019]. Let $\epsilon_t$ denote the value of $\epsilon$ at the $t$-th iteration. We initialize $\epsilon_0$ to a larger value and gradually reduce $\epsilon_t$ to the target $\epsilon$. Specifically, we set $\epsilon_t = \max(\beta \epsilon_{t-1}, \epsilon)$ at the $t$-th iteration, using a scaling constant $\beta \in (0, 1)$. We use $\epsilon_0 = 4$ throughout our experiments, $\epsilon = 0.03$ and $\beta = 0.7$ for text and vision models and $\epsilon = 1.0$ and $\beta = 0.85$ for speech models. Using a larger number of iterations leads to better convergence but we found $T = 20$ sufficient for our experiments.

## C.3   Overhead of soft top-$k$ iterations

The soft top-$k$ iterations are performed for every routed Transformer layer. Although this seems computationally expensive, the actual overhead is very small compared to the overall decoding latency. The complexity only scales linearly with the number of layers and the sequence length, and does not depend on the model dimension $d$. Table 12 showcases the latency numbers on the BoolQ and XSum datasets, when performing batched decoding using a single TPUv4 chip. We observe that the cost of iterations is less than 2% of the total decoding latency. Moreover, the relative cost decreases dramatically as the model size increases, since it does not depend on the model dimension.

## C.4   Additional discussion

This iterative algorithm is closely related to the Sinkhorn algorithm of Optimal Transport (OT). Specifically, the Sinkhorn algorithm solves the entropy-regularized version of Optimal Transport [Cuturi, 2013]. However, our work concerns an different optimization instance. While OT solves a transportation problem where the solution space is defined with the marginal constraints over the rows and columns of a transportation matrix, our optimization problem is constrained with a total budget ($\sum_i \lambda_i = k$) and upper bounds ($\lambda_i \leq 1 \; \forall i$). This leads to different iterative updates.

| Task | Model size | Iteration latency (ms) | Total latency (ms) |
|------|-----------|----------------------|-------------------|
| BoolQ | Base | 0.9 | 48.0 |
| | Large | 1.7 | 105.3 |
| | XL | 1.7 | 297.6 |
| XSum | Base | 1.0 | 513.9 |
| | Large | 2.0 | 1076.2 |
| | XL | 2.0 | 2426.0 |

Table 12: Latency (in milliseconds) of the soft top-$k$ iteration and the total decoding time per batch. We use a single TPUv4 chip and 128 sequences per batch. The maximum iteration overhead is less than 2% of the total latency.

Concurrent to our work, Tai et al. [2022] have used a similar linear program (LP) formulation for soft top-$k$ operation, and have applied the operator for learning sparse neural networks (i.e. model pruning). Compared to our formulation (12), they first reduce the LP to an equivalent instance of optimal transport problem, before introducing the entropy term. As a result, the derived updates are different. In addition, Tai et al. [2022] have introduced an initialization for the dual variables to improve the convergence of their algorithm, whereas we use $\epsilon$ scaling instead. Their implementation can be explored for CoDA as well.

Besides formulating soft top-$k$ using entropy-regularized optimization, there are other possible variants for trainable sparsity. One example is sparsemax [Martins and Astudillo, 2016] that can learn sparse multi-label probabilities. We believe that the sparsemax formulation can generalize from the top-1 to top-$k$ case, but it is beyond the scope of this work. We use the current soft top-$k$ implementation because it is a natural extension of softmax (see discussions in §3), and because it can be solved using simple iterative updates.

## D    Author Contributions

All authors have contributed to running experiments and discussing research ideas. Tao leads the project, developed the conditional architecture, designed the experiments and analyses. Kenton, Yu and Ming-Wei proposed the idea of applying conditional computation for large model adaptation. Joshua demonstrated the conditional architecture is applicable to attention, and implemented the initial version of conditional attention block. Tao, Yanqi, Nan, Vincent, Yuexin, Ming-Wei and Yu conducted the NLP experiments including model pre-training, fine-tuning and various ablation analyses. Siddhartha conducted the majority of the vision experiments. Kenton conducted the vision analysis and advised on the vision experiments. Junwen conducted the majority of the speech experiments. Bo and Yu assisted in trouble-shooting the speech models, ran the model pre-training and provided guidance on the speech experiments. Finally, Tao, Ming-Wei, Junwen and Kenton made the primary contributions to the writing of the paper.

