# OpenReview forum: "Conditional Adapters: Parameter-efficient Transfer Learning with Fast Inference"
_NeurIPS.cc/2023/Conference — NeurIPS 2023 poster_

### Official Review · Reviewer_TS35 · 2023-07-02

**Soundness:** 3 good
**Presentation:** 4 excellent
**Contribution:** 4 excellent
**Rating:** 8
**Confidence:** 4

**Summary:**

The paper proposes Conditional Adapter (CODA), which not only enables parameter-efficient fine-tuning but is also beneficial for the inference speedup. The main idea of CODA is to pass all tokens to a lightweight routing adapter path while selectively passing only certain tokens to the original path. The selection process, named the “Soft Top-K” operation, is conducted by solving an optimization problem that utilizes a generalized entropy function and an iterative searching algorithm. Experimental results on Encoder-Decoder Transformers in NLP, Vision, and Speech applications demonstrate significant inference acceleration with only minimal performance degradation.

**Strengths:**

* The idea of CODA is simple yet powerful. The authors support the effectiveness of the proposed method through sufficient experiments. Although the baseline (Parallel Adapter) achieves higher performance than CODA, I believe the inference speedup can compensate for the performance loss.
* The derivation and implementation of the “Soft Top-K” operation are particularly unique. This operation outperforms simple sigmoid gating or Top-K truncation. Furthermore, this differentiable function has the potential for a broader range of neural network research.


**Weaknesses:**

* Although the authors briefly mentioned CoLT5, it would be better to discuss more the differences between CoLT5 and CODA. The current explanation (L78-79) is somewhat insufficient.

**Questions:**

* How much is the burden of the iterative searching process (T=20) during training? In other words, how does the training time/performance change as iteration steps T are reduced?
* Regarding Figure 5(b), how could the decoding be accelerated? As I understand, CODA does not remove the number of tokens, so the input #tokens to the cross-attention would also not be reduced. Additionally, the readability could be improved if Figure 5 is enlarged in the manuscript (minor issue).
* (Suggestion) I expect CODA can also reduce memory consumption for both training and inference. It would strengthen the paper’s value if there were a comparison regarding memory usage.
* (Suggestion) CODA currently fixes the reduction ratio r for all layers. I understand that it is a simple and structured way, but how about setting different r for different layers? For example, small/large r for low/high layers. This could provide additional insights.


**Limitations:**

The authors have addressed the limitations in Section 7.

---

> ### Author Rebuttal · Authors · 2023-08-09
>
> Thank you for your review and suggestions!
>
> &nbsp;
> **Differences between CoLT5 and CODA:**
>
> We will add a better explanation in the next version of our paper. While both CODA and CoLT5 use conditional activation (token selection) to enhance inference efficiency, the focus of the two works are very different. CoLT5 focuses on optimizing its architecture for very long text (e.g. over 16k tokens), for example, by combining a local attention module with a routed attention module in each layer. The CoLT5 models have to be pre-trained from scratch, and all parameters are fine-tuned on downstream tasks. In comparison, CODA focuses on parameter-efficient transfer learning. CODA models are directly initialized and adapted from an already pretrained dense model, and only the router and small FFN adapters (containing only 5M parameters for example) are trainable in downstream tasks. The strengths of CODA and CoLT5 can be combined for long text tasks, for example, by using local-attention adapters for transfer learning, and by learning CoLT5 models from dense models.
>
> &nbsp;
> ### Answers to other questions:
>
> **1. “How much is the burden of the iterative searching process (T=20) during training? In other words, how does the training time/performance change as iteration steps T are reduced?”**
> A: Please also see our answer to Reviewer 2t3i who asked the same question about soft top-k iteration overhead. The soft top-k iterations (T=20) add about 2-5% overhead during training, and 0.1-2% overhead during inference (decoding). The cost is linear with respect to the number of iterations T. So using T=10 for example will reduce the overhead by another 50%. It is also worth noting that the relative overhead becomes smaller for larger models, since the cost of iterations doesn’t depend on the model dimension.
>
> &nbsp;
> **2. “Regarding Figure 5(b), how could the decoding be accelerated? As I understand, CODA does not remove the number of tokens, so the input #tokens to the cross-attention would also not be reduced. Additionally, the readability could be improved if Figure 5 is enlarged in the manuscript (minor issue).”**
> A: CODA focuses on encoder-heavy tasks such as classification, summarization, question answering and image understanding. The overall decoding latency is dominated by encoding the input sequence. As a result, decoding can be accelerated since CODA makes input encoding significantly faster than the dense baseline. We will improve the readability and add clarity to Figure 5.
> For decoder-heavy tasks, CODA will have limited impact on speed. We listed this limitation in Section 7 and discussed possible future research in our response.
>
> &nbsp;
> **3. “(Suggestion) I expect CODA can also reduce memory consumption for both training and inference. It would strengthen the paper’s value if there were a comparison regarding memory usage.”**
> A: Thank you for your suggestion! You are absolutely correct that CODA indeed reduces memory usage. For example, CODA XL model can decode with a batch size of 512 using a single TPUv4 chip for the summarization task. In contrast, the dense parallel adapter model already runs into out-of-memory (OOM) issues with a batch size of 128.
>
> &nbsp;
> **4. “(Suggestion) CODA currently fixes the reduction ratio r for all layers. I understand that it is a simple and structured way, but how about setting different r for different layers? For example, small/large r for low/high layers. This could provide additional insights.”**
> A: We agree with your suggestion / intuition. We kept a fixed r for simplicity and for an easy, structured implementation of our model. Using (and even learning) different r for different layers is an interesting and promising direction to further improve the accuracy of CODA.

---

> > ### Comment · Reviewer_TS35 · 2023-08-15
> >
> > Thank you for the response - It addressed all of my questions.
> > I believe the CoDA would be a promising direction for efficient fine-tuning domain, and I will keep my score.

---

### Official Review · Reviewer_2t3i · 2023-07-06

**Soundness:** 3 good
**Presentation:** 2 fair
**Contribution:** 3 good
**Rating:** 5
**Confidence:** 3

**Summary:**

This paper proposes a unique modification over parameter-efficient transfer learning by employing a conditional adapter parallel to the pre-trained model. In doing so, a dynamic token selection mechanism is proposed for each block. The selected tokens undergo heavy pre-trained operations, while all the tokens undergo light-weight adapter operations. This greatly improves the inference efficiency of the target dataset. Experiments from language, vision and speech recognition are shown to test the efficacy of the proposed method.

**Strengths:**

1. Wide variety of experiments provided show the superiority of CoDA as compared to parallel-adapter among others.
2. The overall idea of improving the inference speed of large models during the transfer learning stage has huge practical value.
3. A large chuck of questions have been addressed in the main text including - iteration wise performance, the importance of routing, etc.

**Weaknesses:**

1. The presentation of the work can be improved; after section 3.2, the paper seems unorganised, and the scale and location of tables and plots also require some work.
2. Thorough comparison with existing works is missing. There are two streams of comparison possible - [1] Parameter-efficient fine-tuning methods [2] Dynamic/conditional inference methods. Even though experiments are shown on a variety of tasks, it is still unclear how this work compares with existing methods.
3. Final latency with respect to the baseline model is not thoroughly presented, also additional parameters due to parallel branches are not given.

**Questions:**

1. Is there a simple softmax baseline instead of the top-soft-k operation?
2. Is the iterative process for soft top-k done for each forward pass and each layer? If yes, how much overhead does it add?
3. What is the parameter overhead of CoDA and absolute latency numbers for different tasks?
4. There are a variety of standard benchmarks for parameter efficient fine-tuning - VTAB-1k (vision), GLUE (language), among others; why did the authors choose specific benchmarks presented in the paper? If possible, can the authors present results on these standard benchmarks?

**Limitations:**

Yes

---

> ### Author Rebuttal · Authors · 2023-08-09
>
> Thank you for the review and suggestions!
>
> &nbsp;
> **Presentation:**
> We agree that the scale and location of the tables and plots need additional work. Due to the page limit, we have to shrink the size of tables and plots and move some results to the supplementary materials. We will improve the presentation in the next version. Thank you for your comment!
>
> &nbsp;
> **Evaluation and comparisons:**
> We would like to give more context on how we select the baseline. Because there is no prior work on applying conditional computation to adapter / PETL methods, our evaluation focused on comparing CODA against the dense parallel adapter baseline. This baseline is very competitive. Recent work (He et al., 2022; Towards a unified view of parameter-efficient transfer learning) has compared existing PETL methods such as prompt tuning, LoRA, sequential adapters and parallel adapters etc., and found parallel adapters as the best approach. The choice of adapter type in CODA is flexible. Given the reported results from He et al., we implement CODA on top of parallel adapters, and select parallel adapters as our baseline for a direct, apple-to-apple comparison of dense v.s. conditional models.
>
> Please note that we are proposing a new trade-off between accuracy, speed, and parameter efficiency that existing work has not focused on. Our claim is that CODA enables a new “sweet-spot” along these dimensions. We choose to use parallel adapters because it was shown to be the state of the art at the time. It is possible that using CODA with other adapter / PETL methods would lead to higher accuracy, but that difference would be orthogonal to our contribution. We will clarify our choice of baseline in the paper, and are more than happy to hear your suggestions!
>
> &nbsp;
> **Latency and trainable parameters:**
> Our current version of the paper only reports speed and trainable parameters in Table 4, Figure 5 and in the supplementary materials. We will report additional numbers in other tables (e.g. Table 5 and 6). Thank you!
> Please also see our answers to your questions related to latency and trainable parameters.
>
> &nbsp;
> ### Answers to other questions:
>
> **1. “Is there a simple softmax baseline instead of the top-soft-k operation?”**
> A: We implemented and trained a softmax baseline for the rebuttal. This baseline performs much worse than the soft-top-k version. We believe using softmax normalization has an issue as the total contribution (i.e. the mask weight) of all selected tokens is at most 1. Each selected token will receive a small weight, and the output distribution will be shifted (scaled down) and be sensitive to the change of sequence length.
>
> The table below compares softmax v.s. soft top-k methods for a Large model:
>
> | Method | Reduction r | MNLI | RTE | BoolQ |
> | --- | :---: | :---: | :---: | :---: |
> | Soft top-k | 3 | 90.2 | 85.9 | 85.1 |
> | Softmax | 3 | 87.2 | 71.8 | 66.8 |
> | Soft top-k | 5 | 89.7 | 85.2 | 84.3 |
> | Softmax | 5 | 84.7 | 70.4 | 66.8 |
>
> &nbsp;
> **2. “Is the iterative process for soft top-k done for each forward pass and each layer? If yes, how much overhead does it add?”**
> A: Yes, the soft top-k iterations are done for each forward pass and each layer. The overhead of the soft top-k iterations is very small compared to the total decoding latency. Specifically, the complexity is O(TLN) which does not depend on the model dimension d (T is the number of iterations, L is the sequence length and N is the number of layers).
>
> The table below showcases the latency (in milliseconds) of CODA and soft top-k iterations, when decoding using a single TPUv4 core and a batch size of 128. The maximum overhead is less than 2% of the total latency, and the relative overhead decreases dramatically as the model size increases.
>
> | Model size | Task | Reduction r | Total latency (ms) | Soft top-k iteration latency (ms) |
> | --- | :---: | :---: | :---: | :---: |
> | CODA Base  | BoolQ | 5 | 48.0 | 0.9 |
> | CODA Base | BoolQ | 3 | 68.0 | 0.9 |
> | CODA Large | BoolQ | 5 | 105.3 | 1.7 |
> | CODA Large | BoolQ | 3 | 164.6 | 1.7 |
> | CODA XL | BoolQ | 5 | 297.6 | 1.7 |
> | CODA XL | BoolQ | 3 | 466.2 | 1.7 |
> | CODA Base  | Xsum | 5 | 513.9 | 1.0 |
> | CODA Base | Xsum | 3 | 736.3 | 1.0 |
> | CODA Large | Xsum | 5 | 1076.2 | 2.0 |
> | CODA Large | Xsum | 3 | 1534.1 | 2.0 |
> | CODA XL | Xsum | 5 | 2426.0 | 2.0 |
> | CODA XL | Xsum | 3 | 3612.5 | 2.0 |
>
> &nbsp;
> **3. “What is the parameter overhead of CoDA and absolute latency numbers for different tasks?”**
> A: The parameter overhead of CODA is 2M for Base size, 5M for Large size and 10M for XL size. The parallel adapter baseline uses the same amount of trainable parameters. We report these numbers in Table 9 in the supplementary materials. Please see the absolute latency numbers in the table above.
>
> &nbsp;
> **4. “Why did the authors choose specific benchmarks presented in the paper? If possible, can the authors present results on these standard benchmarks?”**
> A: The 4 language understanding tasks we used are indeed standard tasks that are selected in the SuperGLUE and GLUE benchmark. We additionally add SQuAD and XSum to evaluate on generation tasks. Both SQuAD and XSum are commonly used benchmarks as well. We did not evaluate on every single task in GLUE and SuperGLUE, as we have many model variants to evaluate (3 model sizes x 2 reduction rates x 2-4 routing variants).

---

> > ### Comment · Reviewer_2t3i · 2023-08-10
> >
> > The authors have answered a majority of my concerns. Although it would be of greater interest to the community if comparisons were presented as mentioned in Weakness-2. The authors argue that they present a new trade-off between accuracy, speed, and parameter efficiency. Still, it is very specific to parallel adapters, and with each new PETL method would require major engineering. If the authors would give experimental results and comparisons with existing works, it would make the work much stronger.

---

> > > ### Author Response · Authors · 2023-08-19
> > > **Additional experimental results**
> > >
> > > We thank the reviewer for the suggestion. Although we argue our contribution is orthogonal to existing works that focus on improving accuracy, we agree with the reviewer that providing additional comparisons would strengthen our work.
> > >
> > > To this end, we implemented Low-Rank Adapter (LoRA), another competitive PETL method that recently became the most popular choice for LLMs. We followed the very recent development in the QLoRA paper [[Dettmers et al., 2023](https://arxiv.org/pdf/2305.14314.pdf)], to ensure our implementation achieves very strong accuracy. Specifically, QLoRA paper suggests adding low-rank adapters to every linear projection matrices in the Transformer layers to get the best fine-tuning performance (whereas the original work only applies adapters to the query and key projections).
> > >
> > > In order to show that CODA is orthogonal to existing works such as LoRA, we also implemented a CODA variant that applies LoRA instead of the parallel adapters for downstream task finetuning. This variant simply removes the adapter branch and instead adds low-rank adapters to the projection matrices of the pretrained Transformer layers.
> > >
> > > The table below shows the finetuning results using T5 v1.1 Base and Large models:
> > >
> > > | Method | Reduction r | Base MNLI | Base RTE | Base BoolQ |  Large MNLI | Large RTE | Large BoolQ |  Avg on 6 |
> > > | --- | :---: | :---: | :---: | :---: | :---: | :---: | :---: | :---: |
> > > | Parallel Adapter | - | 87.1 | 71.5 | 77.9 | 90.3 | 84.8 | 85.8 | **82.9** |
> > > | CODA (w/ PA) | 3 | 86.6 | 72.6 | 76.6 | 90.2 | 85.9 | 85.1 | **82.8** |
> > > | CODA (w/ PA) | 5 | 86.0 | 70.8 | 76.0 | 89.7 | 85.2 | 84.3 | **82.0** |
> > > | LoRA | - | 88.0 | 73.7 | 80.3 | 90.7 | 85.2 | 86.3 | **84.0** |
> > > | CODA (w/ LoRA) | 3 | 86.2 | 76.9 | 78.4 | 90.3 | 86.3 | 85.8 | **84.0** |
> > > | CODA (w/ LoRA) | 5 | 86.0 | 76.9 | 78.3 | 89.8 | 86.3 | 84.7 | **83.7** |
> > >
> > >
> > > To summarize the observations:
> > >   - The new LoRA baseline achieves stronger accuracy than the Parallel Adapter baseline (84.0 v.s. 82.9 on average), highlighting the effectiveness of recent development on LoRA.
> > >   - The CODA variant using LoRA still achieves very close accuracy compared to its dense counterpart (84.0 v.s. 84.0 / 83.7 on average).
> > >
> > > We believe the additional results strengthen our claims -- that CODA enables a strong trade-off between accuracy and efficiency using conditional activation, and this technique is orthogonal and can be combined with other developments in PETL.

---

### Official Review · Reviewer_XqSH · 2023-07-06

**Soundness:** 3 good
**Presentation:** 3 good
**Contribution:** 2 fair
**Rating:** 5
**Confidence:** 4

**Summary:**

This paper proposed Conditional Adapter (CODA), which can achieve good trade-offs between speed and accuracy. CODA achieves 2x to 8x inference speed-up compared to the state-of-the-art Adapter approach while maintaining moderate to no accuracy loss and the same parameter efficiency across language, vision, and speech tasks.

**Strengths:**

1. The proposed method can greatly speed up inference, while keeping competitive performance.
2. The writing of the paper is clear and easy to understand.
3. The experiments are extensive, which can also well support the arguments of this paper.

**Weaknesses:**

1.  This paper proposes a new adapter and claims to accelerate model inference. However,  there is no direct relationship between the adapter and the inference acceleration. The token pruning approach appears more like an independent module. Therefore,  this paper is more like a combination of token pruning and traditional adapters, and both approaches are not new in the community.
2. The token pruning  is not new, as there have been many methods applied to ViT. Some methods can even achieve inference acceleration without fine-tuning. The author should disscuss the differences with these methods.

Token Merging: Your ViT But Faster, ICLR, 2023
Dynamicvit: Efficient vision transformers with dynamic token sparsification. NeurIPS, 2021
A-ViT: Adaptive tokens for efficient vision transformer, CVPR, 2022

3. Based on 1, I'm not quite clear whether this article proposes an adapter method or a pruning method. From the author's descriptions, i lean towards the former. However, in the experimental section, the author lacks a comparison with newly proposed  parameter-efficient transferring learning (PETL) methods, such as   AdaMix, MAM adapter, and so on.

Adamix: Mixture-of-adapter for parameter-efficient tuning of large language models, EMNLP, 2022
TOWARDS A UNIFIED VIEW OF PARAMETER-EFFICIENT TRANSFER LEARNING, ICLR, 2022

4. It is unclear whether this method can be applied to the currently popular autoregressive models.

**Questions:**

See weakness

**Limitations:**

See weakness

---

> ### Author Rebuttal · Authors · 2023-08-09
>
> Thank you for your review and comments! Please see our response to the weakness points.
>
>
> &nbsp;
> **Contributions:**
>
> One of our major contributions is the insight that the combination of adapters and token pruning is a natural fit. While token pruning is effective for faster inference, it is a destructive process that must be compensated for. Adapter methods introduce a minimal addition that can approximately repair the damage from token pruning. Therefore, our evaluation isn’t more accurate adapters, but having a better trade-off between accuracy, efficiency, and tuned parameters for transfer learning.
>
> We believe our contributions are the following:
>  - We are the first to apply conditional activation to adapter (and PETL) methods to obtain inference efficiency in addition to parameter efficiency.
>  - We provide an evaluation on three modalities, validating and highlighting the general applicability of our method.
>  - We designed an effective algorithm to improve learning efficiency and model quality, including the soft top-k router and up-training CODA from existing dense models.
>
> We will discuss the differences with existing token pruning methods in the paper. Some differences include:
>  - The suggested token pruning works focus on vision tasks, while CODA is applied on text, speech and vision domains. For instance, CODA is shown to preserve the state-of-the-art WER and accelerate inference on the mainstream speech recognition benchmark.
>  - Token merging and top token selection are built with different inductive bias and intuition. Token merging leverages redundancies in visual tokens, while token selection assumes there is a spike of relevance. That is, only a fraction of tokens are most necessary for the prediction task. We believe the latter is more suited for tasks such as question answering, where the information is dense but only a subset of input is relevant to the given question.
>  - CODA introduces a new routing scheme. In previous token dropping works such as DynamicViT and ToMe, if a token is dropped (merged), it will never be reused in subsequent blocks. In contrast, CODA dynamically updates the token representations. Even if the token is not selected for updating in one transformer block, it may still get updated in the later blocks. In text and speech tasks, different tokens might play important roles in different layers. CODA can avoid dropping tokens completely for such scenarios.
>
>
> &nbsp;
> **Evaluation and comparisons:**
>
> Thank you for suggesting the PETL methods!
>
> Please note that our evaluation and choice of baseline are indeed heavily inspired by the suggested work (He et al., 2022; Towards a unified view of parameter-efficient transfer learning). Specifically, He et al. has unified and compared existing PETL methods such as prompt tuning, LoRA, sequential adapters and parallel adapters etc., and found parallel adapters as the best approach. The choice of adapter type in CODA is flexible. Given the reported results from He et al.,  we implement CODA on top of parallel adapters, and select parallel adapters as our baseline for a direct apple-to-apple comparison of dense v.s. conditional models. We will clarify this choice in the paper.
>
> Moreover, we’d like to emphasize that we are proposing a new trade-off between accuracy, speed, and parameter efficiency that existing work has not focused on. Our claim is that CODA enables a new “sweet-spot” along these dimensions. It is possible that using CODA with other adapter / PETL methods (e.g. AdaMix and MAM that use a mixture of multiple adapters) would lead to higher accuracy, but that difference would be orthogonal to our contribution.
>
>
> &nbsp;
> **Limited applicability to auto-regressive decoder:**
>
> We agree with the reviewer and have mentioned this as our limitation in Section 7. CODA selects tokens in a given sequence, but during autoregressive decoding the output sequences are generated on the fly.
>
> Although CODA’s token selection does not apply to autoregressive decoders, the general algorithm of using a light-weight adapter and learning to activate the pretrained model can be useful for future work. While we cannot select tokens along the sequence axis, the soft top-k router is generic and as a result other selection mechanisms can be explored for decoder models, for example:
>  - By selecting tokens within the decoding batch
>  - By selecting parameters or sub-layers of the pretrained model for decoding
>
> These are future work we are working on.

---

> > ### Comment · Reviewer_XqSH · 2023-08-21
> >
> > Thanks for the feedback. The response has solved  most of my concerns.  However, the paper is still limited in some aspects, such as genetilization to auto-regressive model.  In this case, my final score is borderline accept.

---

### Official Review · Reviewer_yqit · 2023-07-12

**Soundness:** 3 good
**Presentation:** 3 good
**Contribution:** 3 good
**Rating:** 5
**Confidence:** 3

**Summary:**

The paper introduces a novel technique called Conditional Adapters (CODA), which aims to enhance transfer learning and improve inference efficiency. CODA surpasses the state-of-the-art Adapter approach by achieving a remarkable 2x to 8x acceleration in inference speed while maintaining moderate accuracy loss and identical parameter efficiency. The paper provides examples of language, vision, and speech tasks where CODA has been extensively evaluated, showcasing the promising results obtained through its application.


**Strengths:**

1. This paper addresses an intriguing problem of accelerating inference speed through efficient parameter tuning.
2. It is commendable that the proposed method is evaluated across three different modalities, indicating its versatility.
3. The model's simplicity and effectiveness are notable. The overall design is straightforward, and the method is technically sound.
4. The paper is well-written and easy to follow.


**Weaknesses:**

1. The availability of the source code is unclear, and it would be beneficial to know if it will be made publicly available for further exploration and reproducibility.
3. Can not be applied to current main-stream auto-regressive models.
2. While the paper covers text and speech modalities, for the visual modality, it would be valuable to benchmark the proposed method on tasks such as image recognition and object detection, in addition to OCR-VQA and Doc-VQA.

**Questions:**

See weakness.

---

> ### Author Rebuttal · Authors · 2023-08-09
>
> Thank you for your review and comments!
>
> &nbsp;
> **Open sourcing and reproducibility:**
>
> We will request to make our code available. Open sourcing the code has to go through our employer’s approval process. In addition, we provide detailed descriptions of our implementation and experimental setup in the supplementary materials for better reproducibility of the work. We will be happy to provide more details if needed.
>
> &nbsp;
> **Limited applicability to auto-regressive decoder:**
>
> We agree with the reviewer and have mentioned this as our limitation in Section 7. CODA selects tokens in a given sequence, but during autoregressive decoding the output sequences are generated on the fly.
>
> Although CODA’s token selection does not apply to autoregressive decoding, the general algorithm of using a light-weight adapter and learning to activate the pretrained model can be useful for future work. While we cannot select tokens along the sequence axis, the soft top-k router is generic and as a result other selection mechanisms can be explored for decoder models, for example:
> - By selecting tokens within the decoding batch
> - By selecting parameters or sub-layers of the pretrained model for decoding
>
> These are future work we are working on.
>
>
> &nbsp;
> **Evaluation on other vision tasks:**
>
> We agree that more traditional vision experiments would be useful. Due to space limitations we decided to focus on the setting that would highlight our strengths. In the case of language understanding within images, the input length is long due to the need for high resolution images. In addition, the sparsity pattern is expected and known a priori because (1) only certain regions of the input image are related to the question, and (2) there is a lot of whitespace (background) in the image. Our VQA experiments can serve as a visualization-friendly sanity check that CODA is able to learn the correct sparsity patterns when they occur in the data, which are indeed observed in Figure 6.

---

### Author Rebuttal · Authors · 2023-08-09

We thank all reviewers for their insightful comments and feedback.

Most reviewers agree that there is an thorough evaluation and analysis in the paper (R1, R2, R3, R4), the tackled problem and proposed technique are important and have a broad practical value (R3, R4), and the paper is very-well presented and written (R1, R2, R4).

The main concerns of the paper include comparison with existing work (R2, R3), and the applicability to auto-regressive decoding (R1, R2).


&nbsp;
**1. Evaluation and comparisons with existing work:**

We’d like to emphasize that we are proposing a new trade-off between accuracy, speed, and parameter efficiency that existing work has not focused on. CODA is not intended / claimed to be the most accurate adapter (which recent work has made a lot of contributions on). Our claim is that CODA enables a new “sweet-spot” along these dimensions -- that is, adding conditional activation to adapter methods offers a nice trade-off between accuracy and efficiency.

We selected parallel adapter as our baseline because it is reported to be the strongest among many PETL methods (e.g. prompt tuning, LoRA, sequential adapters and parallel adapters) in (He et a., 2022; towards a unified view of parameter efficient transfer learning). It is possible that using CODA with newly proposed adapter / PETL methods (e.g. AdaMix and MAM that use a mixture of multiple adapters) would lead to higher accuracy, but that difference would be orthogonal to our contribution.

&nbsp;
**2. Limited applicability to auto-regressive decoding:**

We agree with the reviewers and have discussed this as our limitation in Section 7. CODA selects tokens in a given sequence, but during autoregressive decoding the output sequences are generated on the fly (and therefore not immediately available).

Although CODA is not directly applicable to autoregressive decoder, the general algorithm of using a light-weight adapter and learning to activate the pretrained model can be useful for future work. While we cannot select tokens along the sequence axis, the soft top-k router is generic (R4) and as a result other selection mechanisms can be explored for decoder models, for example:
 - By selecting tokens within the decoding batch
 - By selecting parameters or sub-layers of the pretrained model for decoding

These are future work we are working on.

&nbsp;
**3. Answers to additional questions:**

In addition to these concerns, reviewers suggested us to compare with another softmax routing variant, report the overhead of soft top-k iterations and also discuss the differences with other token dropping works. We provided the detailed answers to these questions / suggestions in the separate response to each reviewer. We thank the reviewers again and are more than happy to hear additional suggestions.

---

### Comment · Area_Chair_eMYn · 2023-08-19
**Discussions are required for the submission**

Dear all reviewers，

Thank you very much for your great efforts in reviewing the referred submission. Now the authors have provided responses regarding your concerns. Would you please read the authors response and see whether your concerns have been addressed or not. You are welcome to raise further concerns if necessary so that the authors can respond to them timely. Your great service would be very important for the community to make final decisions.

Best regards，
Your AC

---

### Decision · Program_Chairs · 2023-09-21

**Decision:**

Accept (poster)

**Comment:**

This paper proposes a unique modification over parameter-efficient transfer learning by employing a conditional adapter parallel to the pre-trained model. The paper introduces a novel technique called Conditional Adapters (CODA) to enhance transfer learning and improve inference efficiency. CODA surpasses the state-of-the-art Adapter approach by achieving a remarkable 2x to 8x acceleration in inference speed while maintaining moderate accuracy loss and identical parameter efficiency. The paper provides examples of language, vision, and speech tasks where CODA has been extensively evaluated, showcasing the promising results obtained through its application.